# Common Factors in the Term Structure of Credit Spreads and Predicting the Macroeconomy in Japan

**Takeshi Kobayashi** 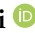

Graduate School of Management, Nagoya University of Commerce and Business, 1-3-1 Nishiki Naka-ku, Nagoya 460-0003, Japan; kobayashi@gsm.nucba.ac.jp

**Abstract:** This study extracts the common factors from firm-based credit spreads of major Japanese corporate bonds and examines the predictive content of the credit spread on the real economy. Instead of employing single-maturity corporate bond spreads, we focus on the entire term structure of the credit spread to predict the business cycle. We extend the dynamic Nelson-Siegel model to allow for both common and firm-specific factors. The results show that the estimated common factors are important drivers of individual credit spreads and have substantial predictive power for future Japanese economic activity. This study contributes to the literature by examining the relationship between firm-based credit spread curves and economic fluctuation and forecasting the business cycle.

**Keywords:** credit spread; common factor; macroeconomic forecast

## 1. Introduction

Predicting the future economy is of great interest to practitioners and policymakers. This study considers whether and how the information contained in the term structure of credit spreads can better forecast the future Japanese economy. Exploring the relationship between credit spreads and future real activity can be motivated by the "financial accelerator" theory developed by Bernanke and Gertler (1989). A key concept in this framework is the external finance premium, the difference between the cost of external funds and the opportunity cost of internal funds due to financial market frictions. The financial accelerator theory argues that a rise in the external finance premium makes outside borrowing costlier, reduces the borrower's spending and production, and consequently restricts aggregate economic activity. Gilchrist and Zakrajšek (2012) show that a reduction in the supply of credit through a deterioration in financial intermediaries' creditworthiness leads to the widening of credit spreads and a subsequent reduction in spending and production. The effectiveness of credit spreads as predictors of economic activity has been confirmed empirically (e.g., Gilchrist et al. 2009; Gilchrist and Zakrajšek 2012; Mueller 2009).

While several researchers have provided empirical evidence on the performance of credit spreads as predictors of real activity, there is still an open debate on which credit spread is the best proxy for the external finance premium. Mueller (2009) finds that credit spreads across the entire term structure and for rating categories ranging from AAA to B have predictive content. Gertler and Lown (1999) and Mody and Taylor (2004) argue that the correct measure is long-term high yield spreads, while Chan-Lau and Ivaschenko (2002) support the use of investment grade credit spreads. Gilchrist et al. (2009) and Gilchrist and Zakrajšek (2012) construct the "GZ spread" from firm-level data and conclude that credit spreads have substantial forecasting power for future economic activity. Most empirical analyses of corporate bond spreads in Japan have explained the single-maturity corporate bond spreads index by regressing macroeconomic variables (e.g., Nakashima and Saito 2009). However, the existing Japanese credit index is difficult to analyze because the yield for the rating class to which corporate bonds with huge outstanding amounts belong is significantly biased. Okimoto and Takaoka (2017) combine data drawn from the Japan Securities Dealers Association (hereafter JSDA) with Thomson Reuters Bond

Credit Curve to create the credit curve index. Kobayashi (2017) focuses on Japanese credit spreads at the firm level and shows the goodness of fit of the study model by aggregating firm-based spreads. However, that study fails to disentangle the common and firm-specific factors contained within the firm's credit spread. To overcome this difficulty, this study extracts the common factors of the credit spread from the term structure of the individual Japanese corporate bond spread date. Our method has the advantage of extracting information on investors' view of the economy. To confirm the information content of the estimated common factors, we conduct in-sample and out-of-sample analyses to examine the predictive ability for economic activity. We find that the estimated common factors are important drivers of individual credit spreads and that these factors, especially the slope factor, have substantial predictive power for future Japanese economic activity.

The remainder of this paper is organized as follows. Section 2 demonstrates how to construct firm-based credit spread curve data. Section 3 explains the underlying term structure model. Section 4 presents the estimation results. Section 5 explains the prediction of economic activity, and Section 6 concludes.

## 2. Data

We use over-the-counter bond transactions data provided by JSDA to estimate zero coupon yields via B-spline methods developed by Steeley (1991). The criteria for selecting corporate bonds are as follows:

1. Observation period: Firms whose time series has over six years of data during the period September 1997 to December 2011.
2. Time to maturity: Corporate bonds of different maturities that have at least seven years for each month.
3. Number of prices: A minimum of five prices of bonds for every month.
4. Industry: The electric power sector is excluded to omit spread widening after the Great East Japan Earthquake.

Corporate bond spread is created by subtracting the corporate yield from the same maturity of the government bond yield. Accordingly, the final sample comprises 26 firms: 14 manufacturing and 12 nonmanufacturing firms (Appendix A Table A1). These credit spread dynamics are driven by idiosyncratic factors, while common factors play an important role in determining the shape of the term structure of credit spreads (Figure 1).

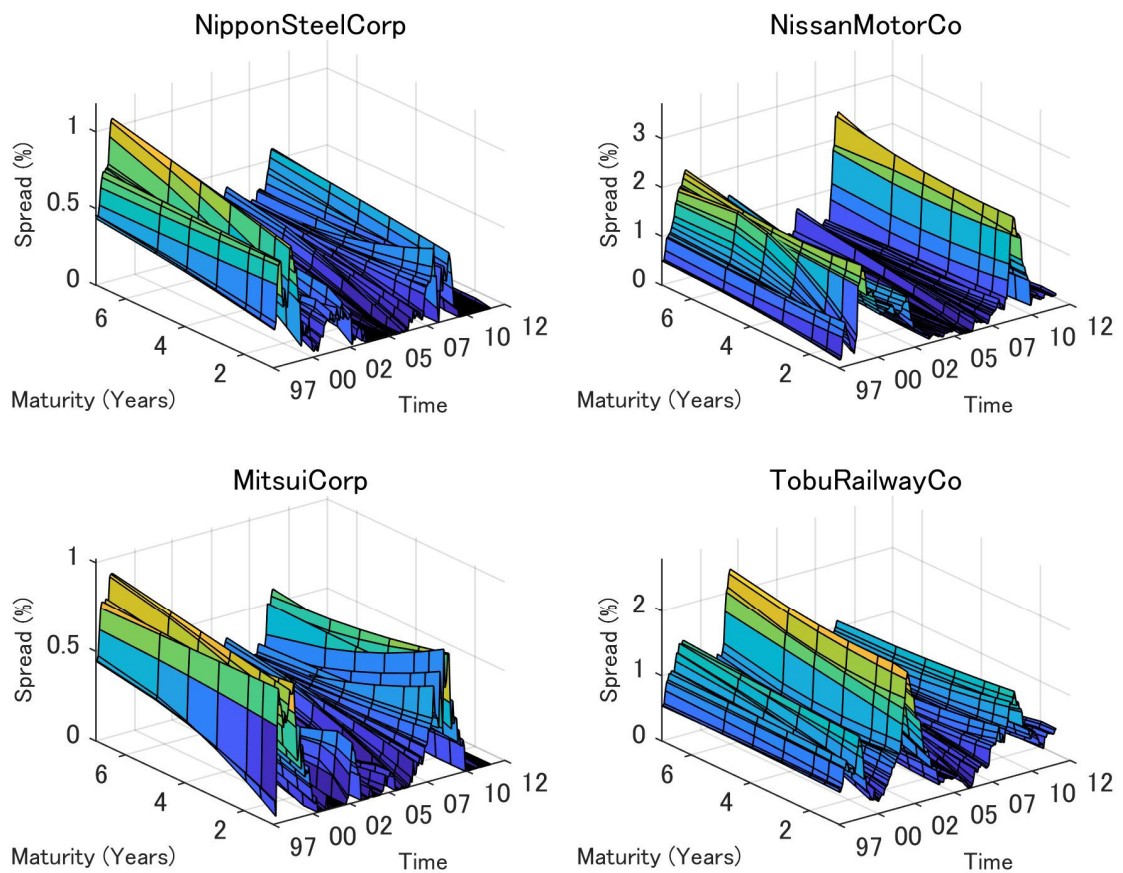

**Figure 1.** Credit spreads across companies and time. Note: Time series of the term structure of credit spreads of 4 of the selected 26 names.

### 3. Model

#### 3.1. Single-Firm Credit Spread Model

We apply the dynamic Nelson and Siegel (1987) model developed by Diebold and Li (2006) to estimate the factors of firm-based credit spread[1].

$$cs_{i,t}(\tau) = l_{i,t} + s_{i,t}\left(\frac{1 - e^{-\lambda\tau}}{\lambda\tau}\right) + c_{i,t}\left(\frac{1 - e^{-\lambda\tau}}{\lambda\tau} - e^{-\lambda\tau}\right) + v_{i,t}(\tau) \qquad (1)$$

where $cs_i(\tau)$ is the zero-coupon yield spread of firm $i$ with $\tau$ months to maturity, and $l_{i,t}$, $s_{i,t}$, $c_{i,t}$, and $\lambda$ are parameters. $v_i(\tau)$ is a disturbance with standard deviation $\sigma_i(\tau)$. $l_{i,t}$, $s_{i,t}$, and $c_{i,t}$ are interpreted as the latent factors of the term structure of credit spread; these indicate level, slope, and curvature factors, respectively.

#### 3.2. Multiple-Firms Credit Spread Model

This section extends the basic model to a multifirm environment following Diebold et al. (2008; henceforth, DLY). The single-firm model may be adapted to an *N*-firm approach. Common credit spreads $CS_t(\tau)$ and the common level and slope, $L_t$ and $S_t$, are not observable[2].

$$CS_t(\tau) = L_t + S_t\left(\frac{1 - e^{-\lambda\tau}}{\lambda\tau}\right) + V_t(\tau) \qquad (2)$$

---

[1]  Abdymomunov et al. (2016) employ the Nelson-Siegel model to estimate the credit spread curve index, while Krishnan et al. (2010) and Kobayashi (2017) use it for the firm-based credit spread curve.

[2]  We focus on the model with the level and slope factors because the estimation of the curvature factor is generally associated with low precision due to missing data in most of the credit spread.

We conduct a principal component analysis (PCA) of the estimated firm-credit spread factors using Diebold and Li's (2006) approach in Figure 2. The PCA shows that about 50% of the variation in the level and slope are driven by the first principal components (henceforth, PC1).

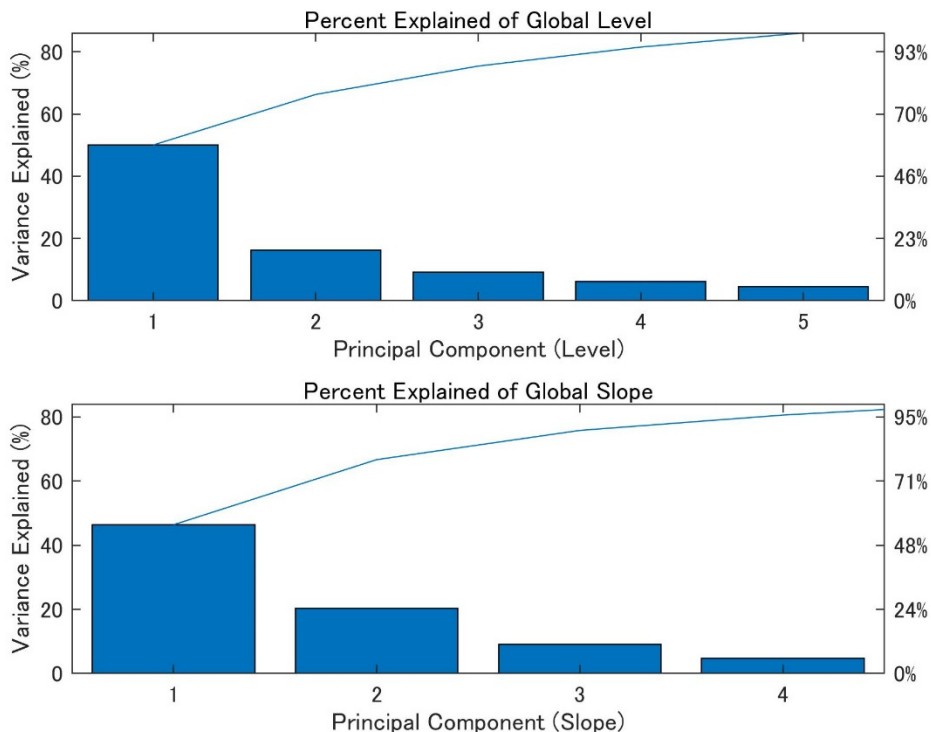

**Figure 2.** Principal Component Analysis.

*3.3. State-Space Representation and Estimation Methods*

These unobservable common factors are common to every firm. They follow the AR (1) model as follows:

$$\begin{pmatrix} L_t \\ S_t \end{pmatrix} = \begin{pmatrix} \Phi_{11} & \Phi_{21} \\ \Phi_{12} & \Phi_{22} \end{pmatrix} \begin{pmatrix} L_{t-1} \\ S_{t-1} \end{pmatrix} + \begin{pmatrix} U_t^l \\ U_t^s \end{pmatrix}, \tag{3}$$

where $\left[\Phi_{ij}\right], i = 1,2$ and $j = 1,2$ is the autoregressive coefficient; $U_t^n$ are disturbances such that $E\left(U_t^n U_t^{n'}\right) = (\sigma^n)^2$ if $t = t'$ and $n = n'$ and 0 otherwise, $n = l, s$. The model then decomposes the firm-specific level $l_{i,t}$ (slope, $s_{i,t}$) into a common level $L_t$ (slope, $S_t$) and some idiosyncratic factor $\varepsilon_{i,t}^n$ whose mean is null:

$$l_{i,t} = \alpha_i^l + \beta_i^l L_t + \varepsilon_{i,t}^l \tag{4}$$

$$s_{i,t} = \alpha_i^s + \beta_i^s L_t + \varepsilon_{i,t}^s, \tag{5}$$

where $\left\{\alpha_i^l, \alpha_i^s\right\}$ are constant terms, $\left\{\beta_i^l, \beta_i^s\right\}$ are loadings on common factors, and $\left\{\varepsilon_{it}^l, \varepsilon_{it}^s\right\}$ are firm idiosyncratic factors, $i = 1 \ldots N$.

Because constant terms are included in Equations (4) and (5), the firm idiosyncratic factors have zero mean. Moreover, because of the magnitudes of the common factors and factor loadings, their disturbances have a unit standard deviation, namely, $\sigma^n = 1, n = l, s$.

$$\begin{pmatrix} \epsilon_{i,t}^l \\ \epsilon_{i,t}^s \end{pmatrix} = \begin{pmatrix} \phi_{i,11} & \phi_{i,21} \\ \phi_{i,12} & \phi_{i,22} \end{pmatrix} \begin{pmatrix} \epsilon_{i,t-1}^l \\ \epsilon_{i,t-1}^s \end{pmatrix} + \begin{pmatrix} \mu_{i,t}^l \\ \mu_{i,t}^s \end{pmatrix} \tag{6}$$

where $\left[\phi_{i,jk}\right], j = 1, 2$ and $k = 1, 2$ is the autoregressive coefficient; $\mu_t^n$ are disturbances such that $E\left(\mu_t^n \mu_t^{n'}\right) = (\sigma^n)^2$ if $t = t'$ and $n = n'$ and 0 otherwise, $n = l, s$. We also assume that $E\left[\mu_{t,t-s}^n U_t^{n'}\right] = O$ for all $n, n', i$ and $s$. In state-space representation, Equations (3) and (6) are state equations. The measurement equation can be represented more compactly by using the following matrix notation:

$$\begin{pmatrix} cs_{i,t}(\tau_1) \\ cs_{i,t}(\tau_2) \\ \dots \\ cs_{N,t}(\tau_J) \end{pmatrix}_{JN \times 1} = A \begin{pmatrix} \alpha_1^l \\ \alpha_1^s \\ \dots \\ \alpha_N^s \end{pmatrix} + B \begin{pmatrix} L_t \\ S_t \end{pmatrix} + A \begin{pmatrix} \epsilon_{i,t}^l \\ \epsilon_{i,t}^s \\ \dots \\ \epsilon_N^s \end{pmatrix} + \begin{pmatrix} v_{i,t}(\tau_1) \\ v_{i,t}(\tau_2) \\ \dots \\ v_{N,t}(\tau_J) \end{pmatrix} \tag{7}$$

where

$N$ is the number of firms;
$J$ is the number of maturities;
$A$ and $B$ are conforming matrices; and
$v_{N,t}(\tau_J)$ are measurement errors.

$$A = \begin{pmatrix} 1 & \frac{1-e^{-\lambda\tau_1}}{\lambda\tau_1} & 0 & \dots & 0 \\ 1 & \frac{1-e^{-\lambda\tau_2}}{\lambda\tau_2} & & & 0 \\ \dots & \dots & \dots & \dots & \dots \\ 0 & 0 & \dots & 1 & \frac{1-e^{-\lambda\tau_2}}{\lambda\tau_2} \end{pmatrix}_{JN \times 2N}$$

$$B = \begin{pmatrix} \beta_1^l & \beta_1^s\left(\frac{1-e^{-\lambda\tau_1}}{\lambda\tau_1}\right) \\ \beta_1^l & \beta_1^s\left(\frac{1-e^{-\lambda\tau_2}}{\lambda\tau_2}\right) \\ \dots & \dots \\ \beta_N^l & \beta_N^s\left(\frac{1-e^{-\lambda\tau_2}}{\lambda\tau_2}\right) \end{pmatrix}_{JN \times 2}.$$

We follow a convenient two-step estimation method proposed by DLY. First, we obtain the latent factors (level and slope) for each firm. Second, we use the previously obtained estimates in Equations (3) and (6) to extract the common factors via the maximum likelihood method.

## 4. Estimation Results

### 4.1. Estimated Common Factors

We show the estimated common factors and PC1 in Figure 3. Close linkage between the common factor and PC1 for firm level and slope is confirmed. The correlation between the common factor and PC1 for firm level is 0.89, while it is 0.92 between the common factor and PC1 for firm slope.

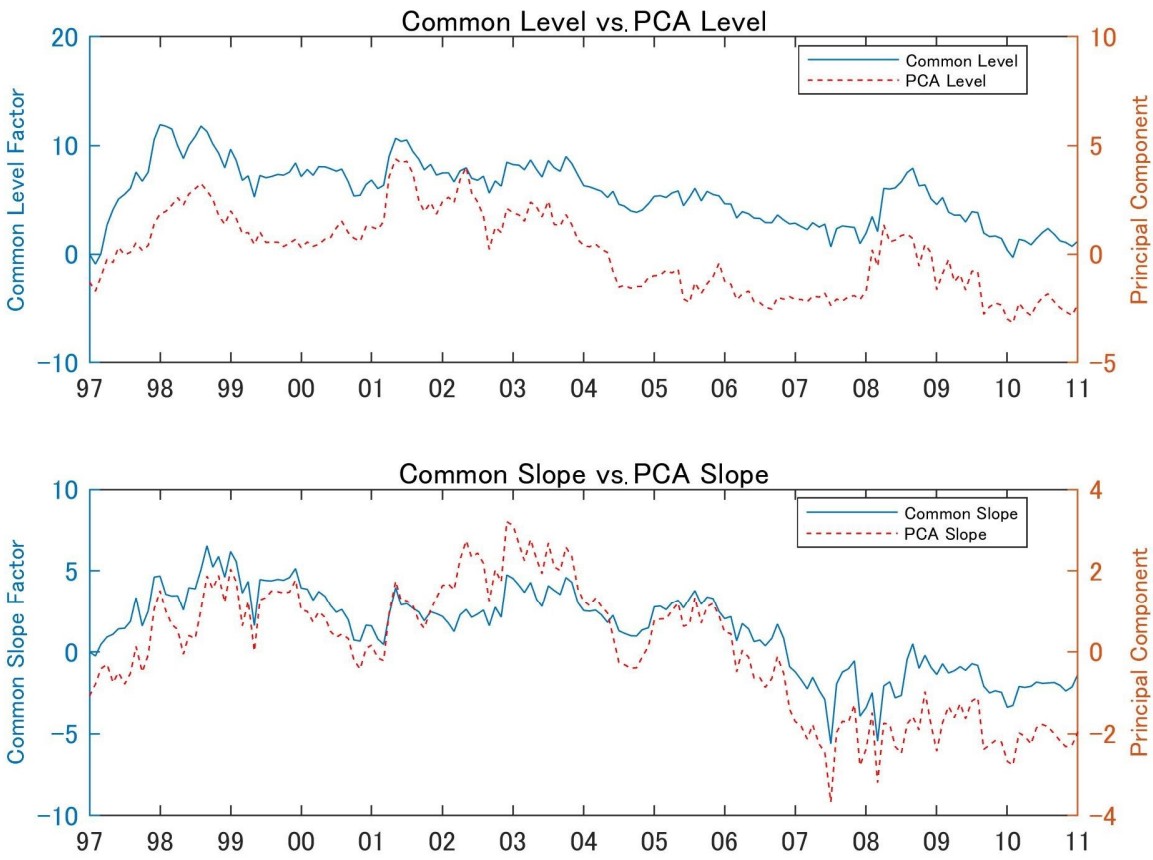

**Figure 3.** Common factor vs. PC1 of firm level and slope. Note: The estimated common factor is indicated by a solid line and PC1 by a dashed line.

### 4.2. Variance Decomposition

A specific firm factor variance can be evaluated as a proportion of the common and idiosyncratic variances. The formulation of firm idiosyncratic factors can be derived from Equations (5) and (6) through a simple definition of variance:

$$\text{var}(l_{i,t}) = \left(\beta_i^l\right)^2 \text{var}(L_t) + \text{var}\left(\varepsilon_{i,t}^l\right) \tag{8}$$

$$\text{var}(s_{i,t}) = (\beta_i^s)^2 \text{var}(S_t) + \text{var}\left(\varepsilon_{i,t}^s\right) \tag{9}$$

Table 1 indicates the results of variance decomposition. The common share is never below 40% and often over 60%, except for a few firms. These findings indicate that the firm-based term structure model can be used to measure risk of the corporate bond portfolio.

**Table 1.** Variance decompositions of the company level and slope factors.

| | Level Factors Volatility | | | | |
|---|---|---|---|---|---|
| | TaiseiCorp | AjinomotoCo | SumitomoChemicalCo | MitsubishiChemicalCorp | JXHoldingsInc |
| Global Factor | 70.2% | 64.5% | 59.8% | 91.4% | 43.0% |
| Idiosyncratic Factor | 29.8% | 35.5% | 40.2% | 8.6% | 57.0% |
| | NipponSteelSumitomoMetalCorp | MitsubishiMaterialsCorp | SumitomoElectricIndustries | NSK | ToshibaCorp |
| Global Factor | 67.1% | 73.4% | 34.6% | 82.5% | 34.3% |
| Idiosyncratic Factor | 32.9% | 26.6% | 65.4% | 17.5% | 65.7% |
| | MitsubishiElectricCorp | Fujitsu | KawasakiHeavyIndustries | NissanMotorCo | ItochuCorp |
| Global Factor | 1.6% | 29.1% | 56.1% | 57.7% | 83.6% |
| Idiosyncratic Factor | 98.4% | 70.9% | 43.9% | 42.3% | 16.4% |
| | MitsuiCorp | MitsubishiEstateCo | TobuRailwayCo | TokyuCorp | EastJapanRailwayCo |
| Global Factor | 58.0% | 43.8% | 78.2% | 83.9% | 59.4% |
| Idiosyncratic Factor | 42.0% | 56.2% | 21.8% | 16.1% | 40.6% |
| | TokyoMetroCo | KintetsuCorp | TokyoGasCo | TohoGasCo | NTT |
| Global Factor | 76.4% | 62.7% | 35.5% | 28.2% | 40.0% |
| Idiosyncratic Factor | 23.6% | 37.3% | 64.5% | 71.8% | 60.0% |
| | KDDICorp | | | | |
| Global Factor | 70.7% | | | | |
| Idiosyncratic Factor | 29.3% | | | | |
| | Slope Factors Volatility | | | | |
| | TaiseiCorp | AjinomotoCo | SumitomoChemicalCo | MitsubishiChemicalCorp | JXHoldingsInc |
| Global Factor | 61.8% | 65.6% | 59.3% | 89.2% | 50.2% |
| Idiosyncratic Factor | 38.2% | 34.4% | 40.7% | 10.8% | 49.8% |
| | NipponSteelSumitomoMetalCorp | MitsubishiMaterialsCorp | SumitomoElectricIndustries | NSK | ToshibaCorp |
| Global Factor | 56.8% | 50.5% | 44.4% | 84.0% | 38.9% |
| Idiosyncratic Factor | 43.2% | 49.5% | 55.6% | 16.0% | 61.1% |
| | MitsubishiElectricCorp | Fujitsu | KawasakiHeavyIndustries | NissanMotorCo | ItochuCorp |
| Global Factor | 0.0% | 42.3% | 54.4% | 54.6% | 52.6% |
| Idiosyncratic Factor | 100.0% | 57.7% | 45.6% | 45.4% | 47.4% |
| | MitsuiCorp | MitsubishiEstateCo | TobuRailwayCo | TokyuCorp | EastJapanRailwayCo |
| Global Factor | 43.2% | 43.6% | 59.4% | 64.2% | 64.2% |
| Idiosyncratic Factor | 56.8% | 56.4% | 40.6% | 35.8% | 35.8% |
| | TokyoMetroCo | KintetsuCorp | TokyoGasCo | TohoGasCo | NTT |
| Global Factor | 43.9% | 56.7% | 0.0% | 17.7% | 55.9% |
| Idiosyncratic Factor | 56.1% | 43.3% | 100.0% | 82.3% | 44.1% |
| | KDDICorp | | | | |
| Global Factor | 13.9% | | | | |
| Idiosyncratic Factor | 86.1% | | | | |

Note: We decompose the variation of the firm level and slope factors into common and firm-specific factors for each firm.

## 5. Predicting Economic Activity

The forecasting performance of the information content of credit spreads for economic activity is examined both in sample and out of sample. We examine the predictive ability by comparing the models including macro and estimated common level and slope factors. We choose the gross domestic product growth (GDP), consumer price index (CPI), and the unemployment rate (UE).

The GDP growth data are taken from the Cabinet Office on a quarterly basis, which are converted using spline interpolation to monthly data. The CPI and UE data are collected from the Ministry of Internal Affairs and Communications.

### 5.1. In-Sample Predictive Power of Credit Spreads

Table 2 shows the in-sample explanatory power of credit spreads for 3-, 6-, 12-, and 24-month forecast horizons. We regress the macro variables $m_t$ on the variables shown in Table 2. When forecasting CPI and UE, the inclusion of credit spreads leads only to a modest improvement in the in-sample fit in the three- to six-month forecast horizons. In contrast, the inclusion of credit spreads in forecasting GDP leads to a substantial increase in predictive accuracy, especially in the six- to twelve-month forecast horizons.

**Table 2.** In-sample predictive content of credit spreads for economic activity.

| Model | Variables | | | GDP | CPI | UE | GDP | CPI | UE |
|---|---|---|---|---|---|---|---|---|---|
| | | | | **Forecast Horizon h = 3 (months)** | | | **Forecast Horizon h = 12 (months)** | | |
| | | | | GDP | CPI | UE | GDP | CPI | UE |
| | | | | Adjusted R-squared | | | Adjusted R-squared | | |
| M1 | Macro | Credit Level | Credit Slope | 0.767 | 0.696 | 0.898 | 0.698 | 0.410 | 0.524 |
| M2 | Macro | Credit Level | | 0.742 | 0.698 | 0.883 | 0.453 | 0.414 | 0.472 |
| M3 | Macro | | | p0.740 | 0.696 | 0.880 | 0.424 | 0.406 | 0.476 |
| | | | | **Forecast Horizon h = 6 (months)** | | | **Forecast Horizon h = 24 (months)** | | |
| | | | | GDP | CPI | UE | GDP | CPI | UE |
| | | | | Adjusted R-squared | | | Adjusted R-squared | | |
| M1 | Macro | Credit Level | Credit Slope | 0.641 | 0.445 | 0.788 | 0.407 | 0.130 | 0.040 |
| M2 | Macro | Credit Level | | 0.540 | 0.441 | 0.748 | 0.363 | 0.000 | -0.010 |
| M3 | Macro | | | 0.519 | 0.445 | 0.747 | 0.336 | 0.005 | -0.010 |

Note: The table reports the adjusted R-squared of ordinary least squares regressions for 3-, 6-,12-, and 24-month forecast horizons.

*5.2. Out-of-Sample Predictive Power of Credit Spreads*

We now examine the predictive content of credit spreads for our three measures of economic activity (GDP, CPI, and UE) using out-of-sample forecasts as shown in Table 3. The state vector follows a vector autoregression (VAR) (1) process:

$$z_{k,t+h} = \mu_z + \Phi_{z,k} z_{k,t} + \epsilon_{z,k,t+h}, \quad \text{where } \epsilon_{z,i} \sim N(0, I) \; for \; k = 1, 2 \tag{10}$$

$$m_{t+h} = \mu_m + \Phi_m m_t + \epsilon_{m,t+h}, \quad \text{where } \epsilon_m \sim N(0, I) \tag{11}$$

where $z_{1,t}$ and $z_{2,t}$ consist of a vector of credit factors and macro variables, $z_{1,t} = (m_t, L_t, S_t)'$ and $z_{2,t} = (m_t, L_t)'$, respectively; $m_t = (GDP_t, CPI_t, UE)'$ is the vector of macroeconomic variables; $\mu$ is the vector of intercept; $\Phi_{z,k}$ and $\Phi_m$ are matrices of the autoregressive process coefficients; and $\epsilon_{z,k,t+h}$ and $\epsilon_{m,t+h}$ are white noise.

For each forecast horizon $h$, we estimate the VAR using 60-month data. We then calculate the (annualized) h-month-ahead economic variables and the associated forecast errors. The forecast data are then updated with an additional month of data; the VAR parameters are re-estimated using this larger observation window and new forecasts are generated. This procedure is repeated through the end of the sample, generating a sequence of out-of-sample forecasts for the three measures of economic activity.

"Ratio" shows that all M1 and M2 values, except CPI at the 6- and 12-month horizons and UE at the 12-month horizon, are below one, which suggests that credit level and slope factors have predictive power for macroeconomic activity.

The results also indicate that the improvement in the credit level factor is relatively small. The inclusion of credit slope factors for GDP and UE leads to a substantial increase in predictive accuracy at the three- to six-month forecast horizon. These improvements are remarkable for GDP at the 12-month forecast horizon. To gauge whether the difference in predictive accuracy between the full model (M1) and the two nested models (M2) and (M3) are statistically significant, a Clark and West (2007) test is conducted. The improvements in predictive accuracy, including in the credit slope factor, are statistically significant at the 10% level except CPI, which indicates the credit slope factor has predictive power for macroeconomic activity.

To assess whether these improvements are due to a specific subperiod, Figure 4 plots the realized values of the out-of-sample forecasts of GDP, CPI, and UE for the 12-month horizon.

**Table 3.** Out-of-sample predictive content of credit spreads for economic activity.

| Forecast Horizon h = 3 (months) | | | | | | | | | | | |
| --- | --- | --- | --- | --- | --- | --- | --- | --- | --- | --- | --- |
| | | | GDP | | | CPI | | | UE | | |
| Model | | Variables | RMSFE | Ratio | CWTest | RMSFE | Ratio | CWTest | RMSFE | Ratio | CWTest |
| M1 | Macro | Credit Level / Credit Slope | 1.320 | 0.866 | | 0.446 | 0.949 | | 0.161 | 0.821 | |
| M2 | Macro | Credit Level | 1.409 | 0.976 | 0.7% | 0.450 | 0.978 | 30.0% | 0.180 | 0.947 | 0.3% |
| M3 | Macro | | 1.429 | - | 2.8% | 0.450 | - | 23.6% | 0.180 | - | 1.5% |
| Forecast Horizon h = 6 (months) | | | | | | | | | | | |
| | | | GDP | | | CPI | | | UE | | |
| Model | | Variables | RMSFE | Ratio | CWTest | RMSFE | Ratio | CWTest | RMSFE | Ratio | CWTest |
| M1 | Macro | Credit Level / Credit Slope | 1.672 | 0.748 | - | 0.585 | 0.935 | - | 0.211 | 0.841 | |
| M2 | Macro | Credit Level | 1.916 | 0.954 | 0.5% | 0.586 | 0.999 | 25.7% | 0.251 | 0.992 | 0.3% |
| M3 | Macro | | 1.961 | - | 1.4% | 0.584 | - | 24.1% | 0.247 | - | 1.1% |
| Forecast Horizon h = 12 (months) | | | | | | | | | | | |
| | | | GDP | | | CPI | | | UE | | |
| Model | | Variables | RMSFE | Ratio | CWTest | RMSFE | Ratio | CWTest | RMSFE | Ratio | CWTest |
| M1 | Macro | Credit Level / Credit Slope | 1.535 | 0.496 | - | 0.636 | 0.947 | - | 0.312 | 0.928 | |
| M2 | Macro | Credit Level | 2.206 | 0.946 | 3.3% | 0.628 | 1.007 | 17.9% | 0.357 | 1.039 | 1.8% |
| M3 | Macro | | 2.257 | - | 2.7% | 0.637 | - | 12.5% | 0.353 | - | 6.7% |
| Forecast Horizon h = 24 (months) | | | | | | | | | | | |
| | | | GDP | | | CPI | | | UE | | |
| Model | | Variables | RMSFE | Ratio | CWTest | RMSFE | Ratio | CWTest | RMSFE | Ratio | CWTest |
| M1 | Macro | Credit Level / Credit Slope | 2.343 | 0.872 | - | 0.841 | 0.873 | | 0.535 | 0.921 | |
| M2 | Macro | Credit Level | 2.446 | 0.944 | 6.0% | 0.919 | 1.005 | 2.2% | 0.553 | 0.993 | 6.9% |
| M3 | Macro | | 2.497 | - | 1.2% | 0.904 | - | 1.4% | 0.550 | - | 5.7% |

Note: Each VAR specification includes a constant and times series of (i) GDP growth, (ii) consumer price index (CPI), (iii) unemployment rate (UE), and (iv) common level and slope factors of the term structure of credit spreads. "RMSFE" indicates the square root of the mean squared forecast error (in annualized percentage points) for each specification. "Ratio" denotes the ratio of the MSFE of the full model, which includes the macro variables, common level, and slope factors of credit spreads (M1), and of the nested model, which includes the macro variables and the common level factor (M2), relative to the MSFE of the nested model including only macroeconomic indicators (M1). "CWTest" denotes the p-value for the Clark and West (2007) test of the null hypothesis that the difference between the MSFE from M1 and the MSFE from M2 or M3 is equal to zero.

The dashed line represents how the forecasts of economic activity using the credit common level and slope factors track the year-on-year growth reasonably well in the actual series during recessionary and expansionary times. This finding indicates that the model incorporating credit slope factors better captures the slowdown in economic activity associated with the 2008–2009 recession relative to the model based on mere credit level and macro indicators.

These findings are in line with those of previous studies on the relationship between credit spread and the predictive ability for macroeconomic activity. For example, Okimoto and Takaoka (2017) suggest that the credit curve of medium-grade corporate bonds has useful information for predicting the business cycle in Japan. The results highlight the usefulness of the term structure of credit spreads in predicting economic fluctuation. Our paper differs from the literature in that we can quantitively show the common factors in individual bond spreads as well as the degree of the contribution of firm-specific factors.

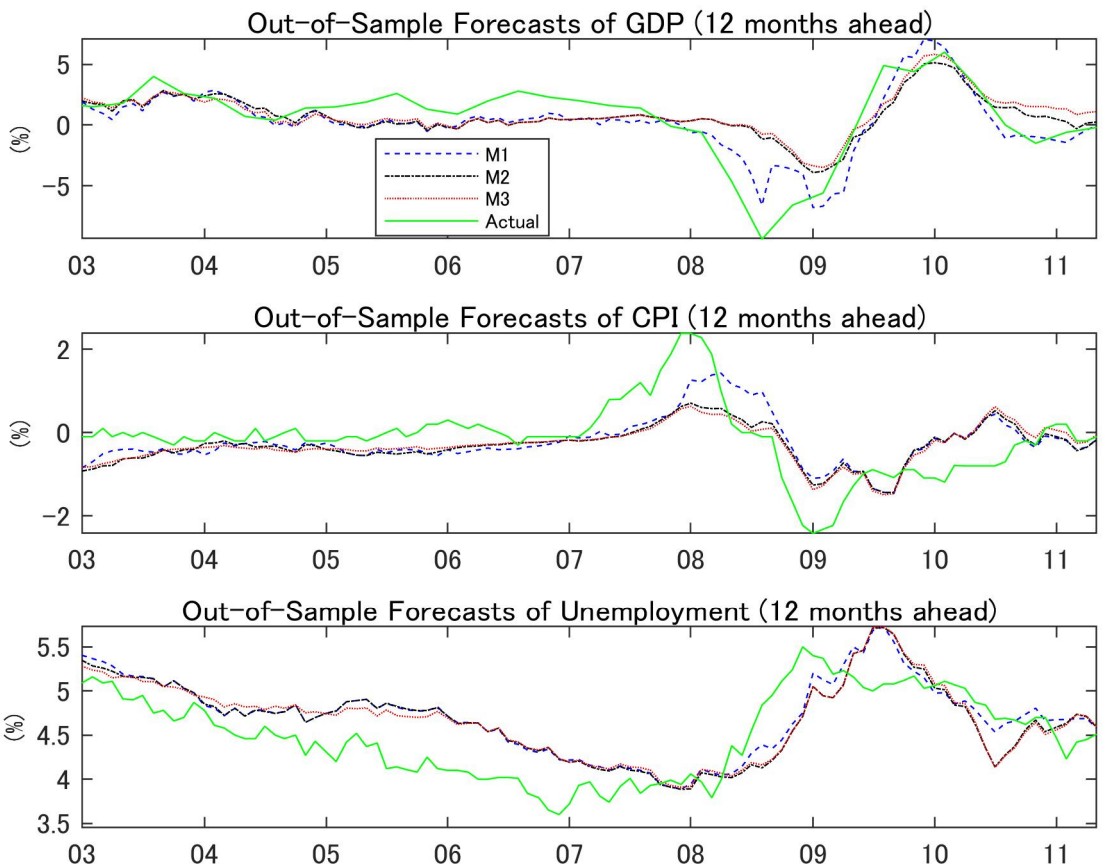

**Figure 4.** Out-of-Sample Forecasts of Economic Activity Indicators. Note: The panels depict out-of-sample forecasts of GDP, CPI, and UE for a 12-month horizon. The figure shows forecasts by M1 (dashed line), M2 (dashed dotted line), M3 (dotted line), and actual data (solid line).

Recent studies focusing on corporate bond spreads in the USA (e.g., Gilchrist et al. 2009; Gilchrist and Zakrajšek 2012), European countries (e.g., Bleaney et al. 2016; Gilchrist and Mojon 2018), and EME (e.g., Caballero et al. 2019) provide strong evidence on their linkage with economic activity. These studies construct credit risk indicators by aggregating the firm-based credit spread. Our approach to estimating the common level and slope factors from the firm-based credit spread should be applicable to various countries and provide a new method of predicting the macroeconomic activity of countries.

## 6. Conclusions

The goal of this study is to extract the common and firm-specific factors from a panel data of individual credit spreads of major Japanese corporate bonds and to examine the predictive content of the credit spread for the real economy. The results indicate that the estimated common factors are important drivers of individual credit spreads, and that these factors, especially the slope factor, have a substantial predictive power for future Japanese economic activity. We can quantitatively show the common factors in individual bond spreads, as well as the degree of the contribution of firm-specific factors, which can be applied for risk management of corporate bond portfolios. These findings provide important insights for discovering useful variables for forecasting Japanese macro variables.

Our approach also provides practical benefits and valuable knowledge for institutional investors, financial executives, and other stakeholders in the corporate bond market. For instance, a firm-based model could be applied directly to risk assessment and corporate bond selection of individual issues for investment managers. Quantifying the proportion of common factors to idiosyncratic firm factors could be useful for calculating the hedge ratio for the common risk factors of corporate bond portfolios. The term structure model can

evaluate the spread level of any term to maturity, which can be used by financial executives in determining the maturity of the bond to be issued.

Further studies capturing nonlinearity and structural change of credit spreads should be undertaken. One possible approach is to consider a Markov-switching dynamic factor model that allows parameters of common factor shifts between regimes.

**Funding:** This research was funded by the JSPS Grant-in-Aid for Scientific Research (grant number 19K34567).

**Data Availability Statement:** Publicly available datasets were analyzed in this study. This data can be found here: the corporate bond and government bond data (https://www.jsda.or.jp/); the gross domestic product (https://www.esri.cao.go.jp/jp/sna/menu.html); the consumer price index (https://www.stat.go.jp/data/cpi/historic.html); the unemployment rate (http://www.stat.go.jp/data/roudou/longtime/03roudou.html).

**Acknowledgments:** I thank Hideyuki Takamizawa, Hajime Tomura, and anonymous referees, as well as participants at the 23th Nippon Finance Association Conference, Society of Applied Economic Time Series Analysis 32th Annual Meeting, Japanese Economic Association 2015 Autumn and 4th Economics & Finance Conference, London, UK for helpful comments.

**Conflicts of Interest:** The author declares no conflict of interest.

## Appendix A

**Table A1.** Breakdown of selected firms.

| # | Name | Industry | # | Name | Industry |
|---|------|----------|---|------|----------|
| No.1 | TaiseiCorp | Construction | No.14 | NissanMotorCo | Assembling |
| No.2 | AjinomotoCo | Primary materials | No.15 | ItochuCorp | Wholesale |
| No.3 | SumitomoChemicalCo | Primary materials | No.16 | MitsuiCorp | Wholesale |
| No.4 | MitsubishiChemicalCorp | Primary materials | No.17 | MitsubishiEstateCo | Real Estate |
| No.5 | JXHoldingsInc | Primary materials | No.18 | TobuRailwayCo | Transportation |
| No.6 | NipponSteeCorp | Primary materials | No.19 | TokyuCorp | Transportation |
| No.7 | MitsubishiMaterialsCorp | Primary materials | No.20 | EastJapanRailwayCo | Transportation |
| No.8 | SumitomoElectricIndustries | Primary materials | No.21 | TokyoMetroCo | Transportation |
| No.9 | NSK | Assembling | No.22 | KintetsuCorp | Transportation |
| No.10 | ToshibaCorp | Assembling | No.23 | TokyoGasCo | Utility |
| No.11 | MitsubishiElectricCorp | Assembling | No.24 | TohoGasCo | Utility |
| No.12 | Fujitsu | Assembling | No.25 | NTT | Telecomminications |
| No.13 | KawasakiHeavyIndustries | Assembling | No.26 | KDDICorp | Telecomminications |

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
