# Peer review of "Common Factors in the Term Structure of Credit Spreads and Predicting the Macroeconomy in Japan"

_ijfs, doi:10.3390/ijfs9020023_

Round 1

Reviewer 1 Report

This is an interesting paper that focuses on the common factors from firm-based credit spreads of major 8 Japanese corporate bonds and examines the predictive content of the credit spread on the real economy. Application of a principal component analysis (PCA) was appropriate. I think the paper is publishable but please proof read the article before final submission. 

Author Response

We thank you for your time in carefully reviewing our paper. Following your comment, we employed the services of an English proofreading company to thoroughly proofread the article.

Reviewer 2 Report

The paper covers an interesting and important issue of the term structure of credit spreads in relation to the real economy indicators, using the example of the Japanese corporate bond market. The overall quality of the paper is high as it is based on sound methodology and the results were interpreted correctly. There are, though, some minor issues that should be improved.

  1. The aim of the study should be stated more clearly both in the abstract and introduction - in its current form it seems rather general.
  2. Author should carefully explain the contribution of the paper and the reasons why it could be of interest for readers from various countries. The study focuses on just one country so it is an important issue to discuss.
  3. Results of the analysis should be compared to the results of the previous studies.
  4. The last section misses some important elements, e.g.:
    • limitations of the analysis,
    • broader implications of the study.

Author Response

We thank you for your time in carefully reviewing our paper. The comments were extremely valuable and helpful in revising the paper. They have led to a significant improvement in the overall quality of the manuscript. The paper has been revised to reflect your suggestions, as follows.

Comment1. The aim of the study should be stated more clearly both in the abstract and introduction - in its current form it seems rather general.

RESPONSE: Following your comment, we have highlighted our approach in the abstract and introduction as follows:

(Abstract, page 1)

Instead of employing single-maturity corporate bond spreads, we focus on the entire term structure of the credit spread to predict the business cycle.

(Introduction, 2nd paragraph, page 2)

Most empirical analyses of corporate bond spreads in Japan have explained the single-maturity corporate bond spreads index by regressing macroeconomic variables (e.g., Nakashima & Saito (2009)).

Our method has the advantage of extracting information on investors’ view of the economy.

Comment2. Author should carefully explain the contribution of the paper and the reasons why it could be of interest for readers from various countries. The study focuses on just one country so it is an important issue to discuss.

RESPONSE:

Following your comment, we have explained the applicability of our study to other countries as follows.

(Section 5, 3rd paragraph, page 11)

Recent studies focusing on corporate bond spreads in the USA (e.g., Gilchrist et al. (2009), Gilchrist and Zakrajˇsek (2012)), European countries (e.g., Bleaney et al. (2016), Gilchrist and Mojon (2018)), and EME (e.g., Caballero et al (2019)) provide strong evidence on their linkage with economic activity. These studies construct credit risk indicators by aggregating the firm-based credit spread. Our approach to estimating the common level and slope factors from the firm-based credit spread should be applicable to various countries and provide a new method of predicting the macroeconomic activity of countries.

Comment 3. Results of the analysis should be compared to the results of the previous studies.

RESPONSE: Following your comment, we compared our work to the results of the previous studies.

(Section 5, 2nd last paragraph, page 11)

For example, Okimoto and Takaoka (2017) suggest that the credit curve of medium-grade corporate bonds has more useful information for predicting the business cycle in Japan.

(Section 5, 3rd paragraph, page 11)

Our paper differs from the literature in that we can quantitively show the common factors in individual bond spreads as well as the degree of the contribution of firm-specific factors.

Comment 4. The last section misses some important elements, e.g.:

limitations of the analysis,

broader implications of the study.

RESPONSE: Following your comment, we have added limitations of the analysis and broader implications of the study

(Section 6, 2nd paragraph, page 11)

Our approach also provides practical benefits and valuable knowledge for institutional investors, financial executives, and other stakeholders in the corporate bond market. For instance, a firm-based model could be applied directly to risk assessment and corporate bond selection of individual issues for investment managers. Quantifying the proportion of common factors to idiosyncratic firm factors could be useful for calculating the hedge ratio for the common risk factors of corporate bond portfolios. The term structure model can evaluate the spread level of any term to maturity, which can be used by financial executives in determining the maturity of the bond to be issued.

(Section 6, 3rd paragraph, page 11)

Further studies capturing nonlinearity and structural change of credit spreads should be undertaken. One possible approach is to consider a Markov-switching dynamic factor model that allows parameters of common factor shifts between regimes.

Reviewer 3 Report

The paper aimed to detect commonalities in credit spreads in Japan. One used these common factors to achieve better forecast accuracy for GDP, CPI, Unemployment.

In general, the paper's idea is simple. However, the literature review does not provide sufficient economic background to support the idea of the paper. The author did not explain why credit spreads are better proxies to explain and forecast a macroeconomic reality.

Author Response

Thank you very much for your comments. The feedback was extremely valuable and helpful in revising the paper. Following your suggestions, we have added a brief discussion about the economic background on why credit spreads are better proxies to explain and forecast a macroeconomic reality.

Comment

The literature review does not provide sufficient economic background to support the idea of the paper. The author did not explain why credit spreads are better proxies to explain and forecast a macroeconomic reality.

RESPONSE:

(Section 1, first paragraph, page 1)

The financial accelerator theory argues that a rise in the external finance premium makes outside borrowing costlier, reduces the borrower's spending and production, and consequently restricts aggregate economic activity. Gilchrist and Zakrajšek (2012) show that a reduction in the supply of credit through a deterioration in financial intermediaries’ creditworthiness leads to the widening of credit spreads and a subsequent reduction in spending and production. The effectiveness of credit spreads as predictors of economic activity has been confirmed empirically (e.g., Gilchrist et al. (2009), Gilchrist and Zakrajšek (2012), Mueller (2009)).